# Differences in Functional Capacity between Oncologic and Non-Oncologic Populations: Reference Values

**DOI:** 10.3390/healthcare12030318

**Published:** 2024-01-25

**Authors:** Egoitz Mojas, Aitor Santisteban, Iker Muñoz-Pérez, Arkaitz Larrinaga-Undabarrena, Maria Soledad Arietaleanizbeaskoa, Nere Mendizabal-Gallastegui, Gonzalo Grandes, Jon Cacicedo, Xabier Río

**Affiliations:** 1Department of Physical Activity and Sport Science, Faculty of Education and Sport, University of Deusto, 48007 Bilbao, Spain; aitorsantisteban@deusto.es (A.S.); iker.munoz@deusto.es (I.M.-P.); a.larrinaga@deusto.es (A.L.-U.); xabier.rio@deusto.es (X.R.); 2Comprehensive Care Group for Patients with Chronic Diseases, Biocruces Bizkaia Health Research Institute, Plaza de Cruces 12, 48903 Barakaldo, Spain; mariasoledad.arietaleanizbeaskoasarabia@osakidetza.eus (M.S.A.); nere.mendizabalgallastegui@osakidetza.eus (N.M.-G.); gonzalo.grandesodriozola@osakidetza.eus (G.G.); 3Radiation Oncology Department, Biocruces Bizkaia Health Research Institute, Cruces University Hospital, Osakidetza, 48903 Barakaldo, Spain; jon.cacicedofernandezdebobadilla@osakidetza.eus; 4Department of Surgery, Radiology and Physical Medicine, University of the Basque Country (UPV/EHU), 48940 Leioa, Spain

**Keywords:** functionality, cancer, handgrip, sit to stand, frailty, muscle mass, intrinsic capacity

## Abstract

This study is focused on the fact that in the context of increasing global aging and cancer diagnoses, additional challenges arise in clinical care. Adequate functionality and body composition are key to coping with antineoplastic treatment, which can lead to better treatment tolerance, survival, and quality of life. This is a cross-sectional comparative study focused on the assessment and comparison of body composition and functionality between cancer patients and a reference population, with the aim of establishing meaningful baseline values. Techniques such as manual dynamometry, the Five-Times Sit-to-Stand test, and bioimpedance were used to collect data from 374 oncologic patients and 1244 reference individuals. The results reveal significant disparities in functionality and body composition among participants, and provide age group-specific adjusted baseline values for those diagnosed with cancer. These findings may have crucial clinical implications for applying particular cut-off points designed for this population group, which makes the assessment process faster and more accurate, enhances the capacity of medical personnel to act quickly, and improves the management of frailty in cancer patients.

## 1. Introduction

The global population is aging rapidly, with an increase in the elderly population worldwide. Estimates indicate that one in six people will be over 60 years of age by 2030 [1,2]. Aging and diseases tend to reduce the physical capacity and functionality of people [3,4], leading to difficulties in activities of daily living and normal functioning in adults and older adults, who are more likely to develop cancer [5].

A challenge in the field of muscle aging is to dissociate the effects of chronological aging on muscle characteristics from the secondary influences of lifestyle and pathological processes [6].

In recent years, there has been a growing interest in understanding the impact of body composition on the occurrence, development and treatment of cancer [7,8]. In elderly patients, age-related changes in body composition, as well as the increased prevalence of obesity, determine the combination of excess weight and reduced muscle mass or strength [9,10,11,12]. Obesity is associated with an increased risk of developing several types of cancer, including breast, colorectal, prostate, and ovarian cancers [13,14].

Adipose and muscle tissues have antagonistic endocrine functions. Thus, when adipose tissue overgrows, the secretion of a series of pro-inflammatory adipokines is promoted along with the inhibition of the secretion of anti-inflammatory adipokines, resulting in low-grade systemic inflammation [15], which weakens the immune system and, in turn, could facilitate the occurrence of cancer [16,17,18]. By contrast, muscle tissue, when contracting, produces another series of cytokines, called myokines, which promote anti-inflammatory mechanisms. These myokines are involved in the autocrine regulation of the metabolism in muscles, as well as in the paracrine/endocrine regulation of other tissues and organs, such as adipose tissue, the liver, and the brain, through their receptors [19].

Therefore, having a good body composition ratio may be paramount to avoiding sarcopenic obesity in addition to many cancer-related pathologies [20,21].

New cases of cancer have progressively increased over the last few years [22]. Preserving muscle strength and health is of vital importance in the oncology patient because cancer treatments can lead to sarcopenia [23,24]. Moreover, as most cancers appear in adulthood, sarcopenia has a high prevalence, not only induced by aging but also by treatments used in cancer [25,26].

Therefore, sarcopenia at the start of oncological therapies has a predictive value for toxicity; consequently, oncologists often reduce the dose and delay cycles, and even interrupt them, leading to a worse prognosis of the disease [23,24].

Furthermore, in patients scheduled for oncologic surgery, sarcopenia has been associated with greater complications in postoperative follow-up along with longer hospitalization days [27], loss of muscle mass and function [28,29], lower tolerance to chemotherapy and radiotherapy [30], and even mortality [31]. Likewise, several studies have shown the efficacy of strength exercise during oncological treatment [7,23,24,32,33], as it can increase muscle mass levels and release natural killer cells [34].

Therefore, the main objective of this study was to evaluate and compare the body composition and functional status of patients diagnosed with cancer to obtain a more complete understanding of the differences in body composition and identify the limitations in the functional capacity of oncologic patients compared to the general population. In addition, another objective of this work was to provide reference values for the variables of hand grip strength and squat test in adults and older adults with cancer in the Basque Country, identifying cut-off points to measure frailty by age group.

## 2. Materials and Methods

### 2.1. Study Design and Tests Used

This is a cross-sectional comparative study in which body values (height, weight, body mass index (BMI), fat percentage, and muscle mass) and functional capacity measured using the Five-Times Sit-to-Stand (5STS) test and manual grip strength (MGS) were collected.

The MGS is recommended in clinical practice and primary care because of its ease of use in the diagnosis of sarcopenic patients [35,36,37]. MGS is one of the tests used as a predictor of low skeletal muscle strength in the diagnosis of sarcopenia [38] and frailty [39], as it is directly correlated with other body regions [40].

The 5STS test has good intra-rater, inter-rater, and test–retest reliability, and is a reliable measurement tool used by experienced or inexperienced raters [41,42]. This test was chosen instead of the Minimum Chair Height Standing Ability Test because it is more effective in patients with osteoarthritis [43], a pathology with a high prevalence of 70% in people over 65 years of age [44].

### 2.2. Participants

The study sample consisted of two well-differentiated groups: on the one hand, patients with cancer, and on the other hand, subjects who acted as a reference group. The age disparity between the two groups is worth mentioning (Table 1).

The subjects with cancer were taken from the database of two projects, Bizi Orain [45] and SEHNeCa [46].

People with any diagnosis of cancer who were currently receiving treatment or who were diagnosed less than two years ago were eligible to participate in a study called Bizi Orain. SEHNeCa is a randomized controlled trial in which patients diagnosed with squamous cell carcinoma of the head and neck receiving curative radiotherapy were recruited. The sample of cancer patients consists of patients after the cancer diagnosis and during treatment [47]. The participants of the reference group were selected using non-probabilistic convenience sampling of the participants of the “Health for the Elderly” program, sponsored by the Bilbao City Council. The inclusion criteria were being 65 years of age or older, being enrolled in the program, and participating voluntarily. The exclusion criteria were inability to walk independently and not having an implanted electronic medical device, such as a pacemaker, as they should not use the bioimpedance scale. Both samples belong to the same geographical region (Vizcaya).

Data collection of the subjects who acted as reference groups was approved by the Ethics Committee of the University of Deusto (reference #ETK-32/18-19) and written informed consent was obtained from each participant before the start of the study. The ethical aspects of the collection of cancer patients were obtained from the Ethics Committee for Research on Medicines of Euskadi (references PI2019016 and PI2020238).

### 2.3. Equipment and Procedure

To measure height, Tanita HR 001 (Tanita Corp., Tokyo, Japan) portable stadiometer was used, and body composition analysis (weight, body fat, and kilograms of muscle) was performed using bioimpedance. In the case of cancer patients, when testing was performed in the laboratory, Inbody 770 (InBody Europe, Amsterdam, The Netherlands) was used. This analyzer is a reliable tool for assessing body composition, with a level of reliability of 98% compared to the results obtained with DEXA [48,49]. Given the difficulty in terms of displacement for the data of the subjects who acted as a reference group, we opted to use a Tanita BC-601 Segment (Tanita Corp., Tokyo, Japan) [39]. Previous research suggests that different types of bioimpedance platforms produce similar results (R^2^ = 0.98) [50].

MGS was obtained using a Camry EH101 (Sensun Weighing Apparatus Group Ltd., Guandong, China) electronic handheld dynamometer, qualified as medical equipment before the Spanish Agency of Medicines and Health Products. The protocol used was in a standing position with the shoulder in slight abduction (approximately 10°), the elbow in full extension, and the forearm and hand in a neutral position [51]. Each person performed the test two times and the higher of the two values was obtained.

For the squat test measurements, a chair (h = 49 cm) and stopwatch were used. Additionally, starting from the 5STS test, the relative and absolute lower-limb powers were calculated [52,53].

The data collection of the reference group was carried out by technicians specialized in physical activity who received specific training from graduates in physical activity and sport sciences in compliance with the test protocols. The evaluations were carried out in the centers and at the times when the participants normally attend the municipal program, with the evaluator being the one who went to the centers.

The measurements of the Bizi Orain and SEHNeCa projects were carried out in the Physiology Laboratory of the University of Deusto by exercise physiologists.

The tests were performed in the following order: first, the height of the participants was recorded and then the analysis of body composition was performed using a bioimpedance meter. This was followed by the 5STS and MGS tests.

### 2.4. Statistical Analysis

The variables analyzed in this study are presented as means and standard deviations. Jamovi (v.2.3.18.0.), R 4.2.2 (R Core Team, 2022), and RStudio (version 2022.12.0.353; Rstudio Team, 2022) were used to analyze the variables. The Levene test was used to evaluate the homogeneity of variance in the data, and the Anderson–Darling, Cramer–von Mises and Kolmogorov–Smirnov tests were used to determine whether the variables were normally distributed. Given the large sample size, parametric tests were chosen because of their statistical power.

In an attempt to predict all dependent variables, multiple regressions were performed, and potential predictors included age group, cancer vs. reference group, and sex, and their interactions (age group × cancer vs. reference group, age group × sex, cancer vs. reference × sex). Stepwise OLS was used to select the final regression model and the explicative variables.

ANOVA tests were established to determine the possible interaction between the different variables and dependent variables. When the ANOVA test showed significant differences between factors, partial eta squared (η^2^) was used as a measure of effect size (ES), using the reference values of small (η^2^ = 0.01), medium (η^2^ = 0.06), and large (η^2^ = 0.14). Subsequently, Tukey’s post hoc test was performed to compare possible differences between factors.

In the comparison, the effect size was calculated using Cohen’s *d* to analyze the standardized mean difference: an effect size of 0.2 to 0.49 was considered small, 0.5 to 0.79 moderate, and 0.8 or higher high [54]. The significance level for all statistical analyses was set at 0.05 (*p* < 0.05).

The P5, P10, P25, P50, P75, P90, and P95 percentiles were chosen as group- and sex-specific reference values.

## 3. Results

A total of 1618 subjects were evaluated: 374 patients (23.1%) were oncologic patients (57.3 years ± 11.00), and 1244 (76.9%) people acted as a reference group (78.1 years ± 5.85), not diagnosed with cancer (Table 1). Of the 1618 participants, 1345 were women (83.1%) and 273 were men (16.9%).

The results reveal statistically significant patterns that highlight the influence and direction of the identified associations, obtaining explanatory values for the different variables studied. There was an interaction between the group to which the participants belonged and their muscle mass (R^2^ = 0.454; *p* < 0.001), as well as between the age of the participants and the result of the 5STS mean r test (R^2^ = 0.80; *p* < 0.001). However, no statistically significant explanatory patterns were found for other variables.

After observing significant differences in all variables between the two groups (Table 2), the sample was segregated according to group, sex, and age (Table 3).

Table 2 highlights the large differences in muscle kg in favor of the reference group (*d* = 2.36). These differences are not reflected in the functional tests, with a low mean difference of *d* = 0.3 in 5STS and a high mean difference of *d* = 1.26 in MGS in favor of cancer patients.

Similarly, numerous significant differences were examined (*p* < 0.05), showing that in subjects with cancer, the kilograms of muscle mass had a strong correlation with the MGS (*r* = 0.77). However, this was not as strongly observed in the reference group (*r* = 0.54). Likewise, a significant correlation was found between absolute lower body power and muscle mass in patients diagnosed with cancer (*r* = 0.75), whereas no correlation was observed in the reference group (*r* = −0.03).

Once the above results were analyzed and differences between gender, the reference group, and the cancer group were observed, it was deemed necessary to calculate the percentiles of the variables measured for the population diagnosed with cancer (Appendix A). In addition, cut-off points were established to assess possible frailty in clinical practice (Table 3).

## 4. Discussion

In the field of oncology and health, it is necessary to understand the differences in the body composition and functionality of patients compared to the non-cancer population. In this way, potential markers and determinants can lead to more effective and personalized interventions aimed at improving patients’ body composition and function, resulting in a higher quality of life.

Several scientific studies have established a significant association between body fat percentage and cancer risk [55,56,57], increasing insulin resistance, inflammation, and altering leptin levels, inducing changes in the metabolism, and causing an increase in the risk of up to 13 different types of tumors [58]. Our results regarding the percentage of body fat in both groups have turned out to be more favorable to the cancer group (32.71 vs. 37.94). This may be due to the difference in mean age between the two groups, as the influence of age on body fat percentage is well documented in the scientific literature [59,60].

A loss of strength and muscle mass in adults has significant implications on the health and quality of life of individuals. From the age of 30 years there is a decline in muscle mass of 3–8% per decade, increasing to 15% from the age of 60 years [61,62,63,64]. Additionally, in oncology patients, this loss is accentuated by various factors, such as chronic inflammation, malnutrition, radiotherapy, and/or chemotherapy treatments, among others [65,66].

As can be seen in the results, cancer patients showed a reduction in muscle mass compared to the reference group (29.0 ± 5.73 vs. 39.3 ± 5.6 kg, *p* < 0.001); however, they obtained higher values in the handgrip (20.7 ± 6.57 vs. 30.3 ± 0.66; *p* < 0.001) and STS (14.7 ± 4.37 vs. 13.4 ± 0.66; *p* < 0.001) strength tests. The prevalence of dynapenia in people older than 65 years is high [67], which may justify these findings. By contrast, muscle strength depends not only on muscle mass [63] but also on the recruitment and activation capacity of motor units at the muscle level [64]. However, the decrease in strength associated with frailty is more pronounced in the lower body [68]. Additionally, leg power has been shown to undergo a more rapid, age-induced decline, which may affect actions such as walking or climbing stairs [69,70].

In addition, STS W-R values are associated with higher levels of frailty and disability, and a poorer quality of life [71]. Therefore, thresholds have been established to determine the need for specific training to improve leg strength and power. These thresholds were established for older adults at 2.6 W-kg^−1^ in men and 2.1 W-kg^−1^ in women [53].

From the obtained values, it can be seen that older adults do not reach the established thresholds. However, cancer patients were at the limit of the thresholds, and we observed that men were slightly below the threshold (2.57 W-kg^−1^) and women were slightly above the threshold (2.17 W-kg^−1^). Therefore, it is advisable for these two populations to perform training aimed at improving lower-limb strength and power.

Similar to other authors, it is necessary to differentiate the lower body power of individuals due to the negative consequences of low values. This strategy can help healthcare professionals more precisely detect possible dysfunction and frailty in cancer patients. Because of its ease of application, the STS W-R test constitutes an implementable and feasible strategy for application in oncologic settings [72].

The results suggest that it is necessary to include an active lifestyle in order to avoid the risks of a sedentary lifestyle, as well as to include strength exercises so as to improve muscle mass and functionality [73,74]. Likewise, in view of the differences in the variables measured between the groups, individualized exercise programs adapted to individual characteristics are recommended [75].

Several factors can interact with the neuromuscular system, such as comorbidities, muscle pathologies, and the use of certain medications. It is important to note that, in this study, no specific data were collected on these factors, which could have a significant impact on the variables analyzed. Although some studies [50,76] have demonstrated a high correlation between bioimpedance platforms, the fact that the same platform was not used makes this a limitation of the study.

Further research is needed to determine the differences that may exist between different types of cancer so that new health-promotion strategies and individualized exercise programs can be designed according to the characteristics of each type of cancer. Likewise, knowing the physical characteristics of the older adult population of a community can lead to the implementation of health-promotion policies that prevent frailty and increase the quality of life.

## 5. Conclusions

In conclusion, cancer patients, as has been shown in several investigations, have less kilograms of muscle; this fact may be due to the disease itself, antineoplastic treatment, or reduced levels of physical activity. However, despite having a lower muscle mass, the participants in the present study within the cancer group had better functional capacity.

Despite the relevance of the current findings for improving frailty detection in healthcare settings, their effective integration into clinical practice remains a challenge. This study highlights the disparity in MGS, kg of muscle mass, and 5STS values between cancer patients and the general population. We propose that the determination of age-specific reference values may be a significant advance in expediting the clinical identification of weaknesses and potential risks in the oncologic population. The application of specific cut-off points tailored to this demographic group facilitates a more accurate and rapid assessment, thereby improving the ability of healthcare professionals to intervene in a timely manner and improve outcomes in the management of frailty in cancer patients.

Likewise, the application of lower-limb power to determine the frailty of the participants is considered suitable. The speed with which this test can be performed and the ease of applying a formula establish an effective and time-saving evaluation method that can also be included in clinical practice by medical specialists or physical exercise professionals. All of this is in order to understand how this variable progresses during the different phases of oncological treatment.

More studies should be carried out on the follow-up, the reference group, and the improvement of muscle mass in oncological patients in order to optimize and individualize treatments more effectively and to achieve better functional capacity and quality of life.

## Figures and Tables

**Table 1 healthcare-12-00318-t001:** Description of the sample.

Group	Age (years)	Height (cm)	Weight (kg)	BMI (kg/m^2^)	Fat (%)	Muscle Mass (kg)	5STS (s)	MGS (kg)
Cancer (N = 374)	57.29 (10.98)	1.65 (0.08)	71.96 (15.57)	26.32 (5.13)	32.71 (9.39)	26.04 (5.73)	13.41 (4.35)	30.27 (10.23)
Reference group (N = 1244)	78.11 (5.85)	1.54 (0.06)	67.2 (10.58)	28.45 (4.05)	37.94 (6.02)	39.33 (5.60)	14.72 (4.36)	20.65 (6.57)

Notes: Data presented as mean (SD); N = population size; BMI = body mass index; 5STS = 5-Times Sit-to-Stand Test; MGS = manual grip strength; manual dynamometry.

**Table 2 healthcare-12-00318-t002:** Descriptive table of the differences between the cancer group and the reference group.

						95% Confidence Interval	
Variables	Mean (SD) Reference Group	Mean (SD) Cancer	Student *t*	*p*-Value	Cohen’s *d*	Inf.	Sup.
BMI (kg/m^2^)	28.5 (4.06)	26.3 (5.14)	−8.35	<0.001	−0.49	−0.61	−0.37
Fat percentage (%)	37.9 (6.02)	32.7 (9.39)	−12.77	<0.001	−0.75	−0.88	−0.62
Muscle mass (kg)	39.3 (5.6)	26 (5.73)	−40.03	<0.001	−2.36	−2.56	−2.15
5STS (s)	14.7 (4.37)	13.4 (4.36)	−5.09	<0.001	−0.30	−0.41	−0.18
5STSAbs (W)	121.91 (44.48)	172.35 (66.72)	16.8	<0.001	0.975	0.86	1.14
5STSRel (W·kg^−1^)	1.80 (0.53)	2.37 (0.66)	16.32	<0.001	0.973	0.87	1.15
MGS (kg)	20.7 (6.57)	30.3 (10.23)	21.52	<0.001	1.26	1.12	1.41

Notes: Data presented as mean (SD); BMI = body mass index; 5STS = 5-Times Sit-to-Stand Test; STSAbs = 5-Times Sit-to-Stand mean power; STSRel = 5-Times Sit-to-Stand power relative; MGS = manual grip strength; Inf. = bottom; Sup. = bottom.

**Table 3 healthcare-12-00318-t003:** Mean values by age and gender of the variables with significant differences in cancer patients.

Age Group and Gender	MGS *		STS *		5STS rel *	
**Women**		**CP**		**CP**		**CP**
G1 = 50–59 (n = 73)	24.3 (5.3) ^3^	19.0	13.2 (3.8) ^4^	9.4	2.2 (0.6) ^2,3,4^	1.6
G2 = 60–69 (n = 58)	22.9 (4.8) ^3^	18.1	14.5 (6.0)	8.5	2.0 (0.5) ^1^	1.5
G3 = 70–79 (n = 18)	19.2 (4.0) ^1,2^	15.2	16.1 (5.0)	11.1	1.7 (0.5) ^1^	1.2
G4 = 80–89 (n = 3)	19.2 (0.4)	-	22.0 (10.4) ^1^	-	1.3 (0.6) ^1^	-
**Men**						
G1 = 50–59 (n = 40)	43.8 (7.9) ^3,4^	35.9	12.3 (4.0) ^3^	8.3	2.9 (0.7) ^3^	2.2
G2 = 60–69 (n = 59)	40.4 (7.1) ^3^	33.3	12.8 (2.8) ^3^	10	2.7 (0.6) ^3^	2.1
G3 = 70–79 (n = 27)	33.4 (5.8) ^1,2^	27.6	15.7 (5.7) ^1,2^	10	2.1 (0.5) ^1,2^	1.6
G4 = 80–89 (n = 2)	29.0 (5.1) ^1^	-	16.6 (6.3)	-	1.9 (0.8)	-
**Women and Men**	**¥**		**¥**		**¥**	
G1 = 50–59 (n = 113)	31.2 (11.3)		12.9 (3.9) ^3,4^		2.5 (0.7) ^3,4^	
G2 = 60–69 (n = 117)	31.7 (10.7)		13.7 (4.7) ^3,4^		2.3 (0.7) ^3^	
G3 = 70–79 (n = 45)	27.7 (8.7)		15.8 (5.4) ^1,2^		2.0 (0.6) ^1,2^	
G4 = 80–89 (n = 5)	23.1 (6.0)		19.8 (8.6) ^1,2^		1.6 (0.7) ^1^	

* Data presented as mean (SD). MGS = manual grip strength, 5STS = 5-Times Sit-to-Stand Test, 5STS rel = relative power, CP = cut-off points values using <1DE by gender and age group; ¥ = significant differences between gender at *p* < 0.001. ^1, 2, 3, 4^ = significant differences at *p* < 0.05 in age subgroups.

## Data Availability

Data supporting the reported results can be obtained by mailing the authors.

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
