# Peer review of "Differences in Functional Capacity between Oncologic and Non-Oncologic Populations: Reference Values"

_healthcare, 2024, doi:10.3390/healthcare12030318_

Round 1
Reviewer 1 Report
Comments and Suggestions for Authors
Thank you for the opportunity to familiarize myself with the content of the article and for the effort put forth by the authors in researching differences in functional capacity between oncologic and nononcologic patients. My suggestions are as follows:
Clarification of Patient Groups:
One of the main points worth specifying is information regarding the oncologic patient group. The authors should precisely state whether the study includes patients undergoing treatment or those who have completed treatment. This detail is crucial as it affects the understanding of results and their interpretation.
Control Group:
It would also be valuable to obtain information about the control group. Did it include healthy individuals without chronic illnesses? A more detailed characterization of the control group would help readers better understand the context of comparisons between oncologic and nononcologic patients.
Conclusions and Proposed Solutions:
In the discussion, there is a lack of reference to the conclusions drawn from the conducted study and proposals for solutions. Authors should consider presenting specific actions resulting from their research, such as the need to differentiate lower body power. Providing information on where and by whom these actions should be taken would enrich the discussion and make the conclusions more practical.
Article Structure:
It's worth paying attention to the structure of the article to ensure it is clear, consistent, and understandable – additional clarification is needed for interpreting the results in Table 2.
Conclusion:
The article contains important information regarding differences in functional capacity between oncologic and nononcologic patients. Clarifying certain aspects and adding specific conclusions and proposed actions would enhance the usefulness of the article for readers.

Author Response
Good morning, thank you very much for your comments.
Please find attached a document with the responses to your comments and the manuscript with change control.
Thank you very much for your time.

Reviewer 2 Report
Comments and Suggestions for Authors
The manuscript shows the differences that oncology and non-oncology patients have in functional capacity.
Study design:
Line 81 to 83: should be placed to another section as justification for example.
Line 85-89: the reliability of the test used should be placed in variables section.
Participants:
Line 91-94: these lines are results.
Inclusion and exclusion criteria must be added.
Time from diagnose, pharmacology or other treatments could be bias in the recruitment of the sample. Have any of these aspects been taken into account?
Lifestyle (habits or daily activity) could also influence the results... are they been considered?
Procedure is not explained: how was developed the valuation? Was the same order followed? Who conducted the evaluations? How many people took part in the evaluations? When and where did they take place?
Is there any researcher blinded?
Variables should be explained better.
Results:
Line 154: the explanation of the abbreviations used in table 1 should be in the table caption, not in the title. Also some abbreviations are missing to be explained
Discussion
Line 231-232: which studies? Please add the references.
The results show differences between men and women but there is no mention in discussion section about it.
More literature that support your results must be added.
Author Response

(The authors gave the same response as above.)

Reviewer 3 Report
Comments and Suggestions for Authors
First of all, congratulations for the work done, then I will mention a number of changes and recommendations in order to obtain clearer and more accurate information.
- Comments on the abstract:
This study is focused should be the correct form. Line 17.
I miss the type of study conducted in the summary.
- Comments on the introduction:
Number under 10 should be written in letters. One in six. Line 31.
Regarding the rest, I consider that in general the introduction is clear and sufficient to introduce the subject. Congratulations.
- Comments on material and methods
- Study design:
Use the same name in the abstract and material and methods, as you use five Times Sit to Stand and 5-squattest/5-repetition chair test respectively.
- Participants:
You should mention the disparity in age between groups.
You should indicate what SEHNeCa is
- Equipment and tests used:
You should have used the same instruments to measure both groups. I know you have stated this as a limitation.
- Statistical analysis:
Statistical significance must be set at p< 0.05 not p<=0.05.
Again, except for these details, I must congratulate the authors for the work done in these sections.
- Comments on results:
Line 151-152. 1618 and 1345 should be 1,618 and 1,345. Check this thorough the manuscript.
Table 1. Check the data of the weight of each group, you have written the same in both, check if it could be a mistake and correct it if necessary.
Table 3. In the table foot you talk about kg of muscle mass and in the table it does not appear.
- Comments on discussion:
STS W-R or STS W R, check this and always use the same.
What about the results of fat mass? You do not discuss them at all and the difference between groups in favor to cancer group should have an explanation.
-General comments:
In general I think it is a good work but the selection of the sample with such age differences in dealing with this subject, complicates the understanding of the results and reduces their validity, since very different age groups are compared.
Author Response

(The authors gave the same response as above.)

Round 2
Reviewer 2 Report
Comments and Suggestions for Authors
Dear authors,
Thank you for following my suggestions.
Reviewer 3 Report
Comments and Suggestions for Authors
All my suggestions have been amended. Good job.